# The Hydraulic Connection Analysis of Dongying Geothermal Fluid Using Hydrochemical Information and Isotope Data in Tianjin Coastal Regions

**Jiulong Liu** [1], **Shuangbao Han** [1,*], **Fengtian Yang** [2] and **Dongdong Yue** [1]

1 Center for Hydrogeology and Environmental Geology Survey, China Geology Survey, Tianjin 300304, China
2 Key Laboratory of Groundwater Resources and Environment, Ministry of Education, Jilin University, Changchun 130021, China
* Correspondence: shuangbaohan@126.com

**Abstract:** Dongying's Paleogene geothermal resources are an important part of the geothermal resources of the Tianjin coastal region. The extent of the geothermal fluid resources and the supply relationship have become increasingly important, and will determine whether demand targets can be met. Dongying's Paleogene formation in the Tianjin coastal regions is widely distributed to the east of the Cangdong fracture, but it is absent west of the Cangdong fracture. On the basis of introducing the geological characteristics and depositional characteristics of the Dongying formation, we analyzed the hydraulic conductivity of the Cangdong fracture to the Dongying formation geothermal reservoir from the aspects of geological condition, dynamic of groundwater level and hydrologic geochemistry. Based on the hydrochemical information and the isotope data gained during the water quality evaluation and isotope data analysis process, we discovered the main chemical composition, hydrogen and oxygen isotope data and geothermal fluid age are significantly different between the Dongying formation geothermal reservoir and the overlying and underlying geothermal reservoirs. It is inferred that the hydraulic conductivity of the Cangdong fracture to the Dongying formation geothermal reservoir in this area is weak, and along the Haihe fracture, where the Haihe fracture intersects with the Cangdong fracture, there is a certain hydraulic conductivity. In addition, there is no obvious hydraulic connection between the Dongying formation and the upper and lower geothermal reservoirs.

**Keywords:** Dongying geothermal fluid; hydraulic connection; hydrochemical information; isotope data; Tianjin coastal regions

## 1. Introduction

The Tianjin Binhai New Area is located in the northern part of the North China plain, near the Bohai Sea. With the development and opening of the Tianjin Binhai New Area, the demand for geothermal energy and other clean energy is increasing [1]. At present, most of the geothermal reservoirs in the area are Minghuazhen formations and Guantao formations of Neogene, which are mainly used for heating and bathing. However, due to over-concentrated exploitation, the head pressure of the geothermal reservoir has dropped, especially in the Guantao geothermal reservoir, which has been seriously overexploited. At present, the exploration resources cannot meet the needs of local development, so new geothermal resources urgently need to be developed. Dongying's geothermal resources are an important part of the geothermal resources of the Tianjin coastal regions [2]. According to the available data, the Dongying formation and its inland lacustrine clastic deposits are widely distributed in the Huanghua depression. The oil logging data show that the porosity of the Dongying formation is more than 20%, and the water-rich property is good. It has been successfully exploited for the first time through perforating and has a certain development potential. Additionally, the exploitation of the rich geothermal resources of the

Tianjin coastal regions will accelerate the long-term goal of carbon neutrality. Geothermal fluids and their supply will determine whether demand targets can be met. Therefore, the study of the Dongying geothermal reservoir has gradually attracted attention.

The Dongying geothermal reservoir is distributed to the east of the Cangdong fracture, and its distribution is wide, but it is absent to the west of the Cangdong fracture [3]. As the main reservoir of this area, the Dongying geothermal reservoir can provide significant back-up energy and serve the economic development of this area. However, the Dongying geothermal reservoir's condition is poor, the unit water inflow of single wells is low, and the understanding and development of Dongying's geothermal resources are still fraught with many problems and difficulties [4]. According to previous work, the Cangdong fracture is a hydraulic conductivity fracture [5]; however, whether the Cangdong fracture is a hydraulic conductivity fracture for Dongying's formation of a geothermal reservoir is a very important question, which is directly related to the potential distribution of Dongying's geothermal resources, the rational arrangement of geothermal wells and the avoidance of development risks. Through the analysis of hydrochemical information, the hydrogen and oxygen isotope data and the stable isotope data between the Dongying formation geothermal reservoir and the overlying and underlying geothermal reservoirs in the Tianjin coastal regions, the important parameters of geothermal hydraulic connection can be obtained in detail, and the method is shown to be effective. This method partly addresses the lack of research into the geothermal fluid circulation of deep geothermal sedimentary aquifers. Therefore, this research involved the deposition environmental analysis of finer sequence stratigraphic units of the Dongying formation [2] and further analyzed its geothermal hydraulic connection, which is of great significance to the appropriate development of Dongying's geothermal resources.

## 2. Materials and Methods

### 2.1. Study Area

The study area is located in the north–central part of the Huanghua depression, which is a grade III tectonic unit. The study area spans two grade IV tectonic units, the Beitang depression and Banqiao depression [3,6,7] (Figure 1). Due to the existence of the Haihe fracture in this area, the current nose-like structure pattern was formed [8]. The nose-like structure zone has a near east–west distribution. Its southern boundary is the Haihe fracture. Its northern boundary is the Tangbei fracture. Its western boundary is the Cangdong fracture, and its eastern side extends into the Bohai Sea and connects to the Xingang semi-anticline. The section inclines of the Haihe fracture and Tangbei fracture are contrary, which leads to the formation of an east–west stripped bedblock in their midst. Its existence affects the sedimentary distribution of a part of the Paleogene strata. The Dongying formation of Paleogene in the Tianjin coastal regions is widely distributed to the east of the Cangdong fracture but is absent to the west of the Cangdong fracture (Figure 2).

The whole length of the Cangdong fracture, from north of Huaizhuang village in the Ninghe district to south of Hulianzhuang in the Jinghai district and into Hebei province, is about 200 km. The length of the northern section in Tianjin is about 120 km, the strike is north-north-east, the dip is south-east-east and the dip angle is 30–48° [9]. The cutting depth is more than 10 km; it cuts through the silicon and magnesium layers of the whole crust, breaks off to the upper mantle, and cuts up to the Minghuazhen formation of Neogene [10]. It controls the development of Mesozoic and Cenozoic basins and forms the boundary between the Cangxian bulge and Huanghua depression [11]. The west plate of the Cangdong fracture is the Cangxian bulge, and the basement roof of the Cangxian bulge is mainly Paleozoic and Meso-neoproterozoic, the buried depth of the bedrock roof is 800~150 m and the upper part is Paleogene of the Cenozoic, which is almost absent. The east plate of the Cangdong fracture is the Huanghua depression. The bedrock roof of the Huanghua depression is mainly Mesozoic and Late Paleozoic. The buried depth of the Dongying formation in the east plate of the Cangdong fracture is similar to the Mesozoic, Cretaceous and Cambrian formations in the west plate of the Cangdong fracture.

The Dongying formation near the Cangdong fracture is relatively thin, and the formation conditions for making up and draining the Dongying geothermal reservoir through the Cangdong fracture are poor.

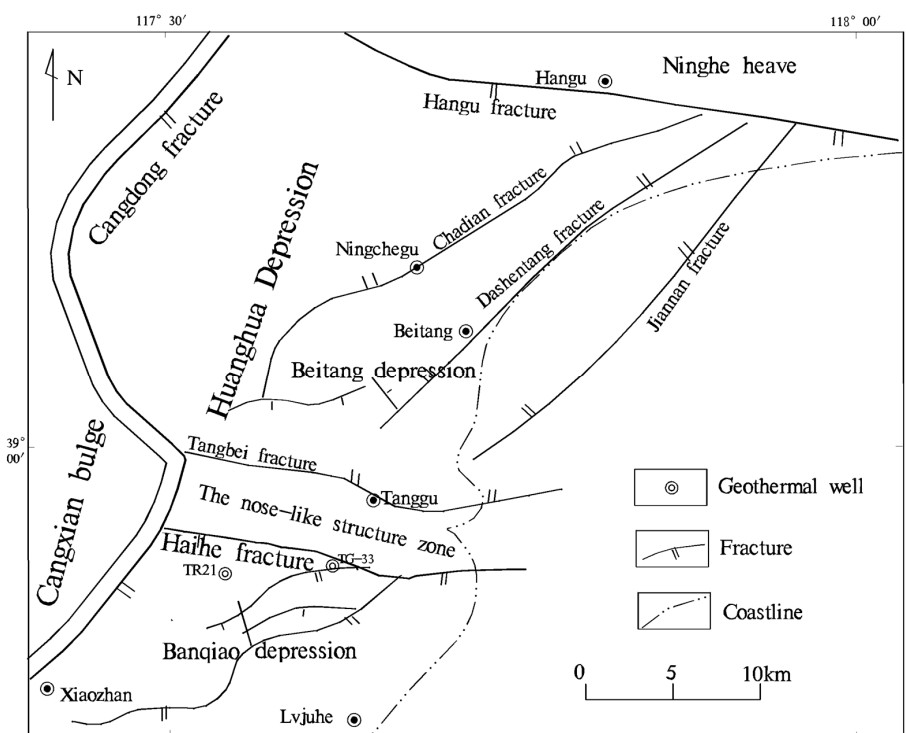

**Figure 1.** Geological structure characteristics of the study area.

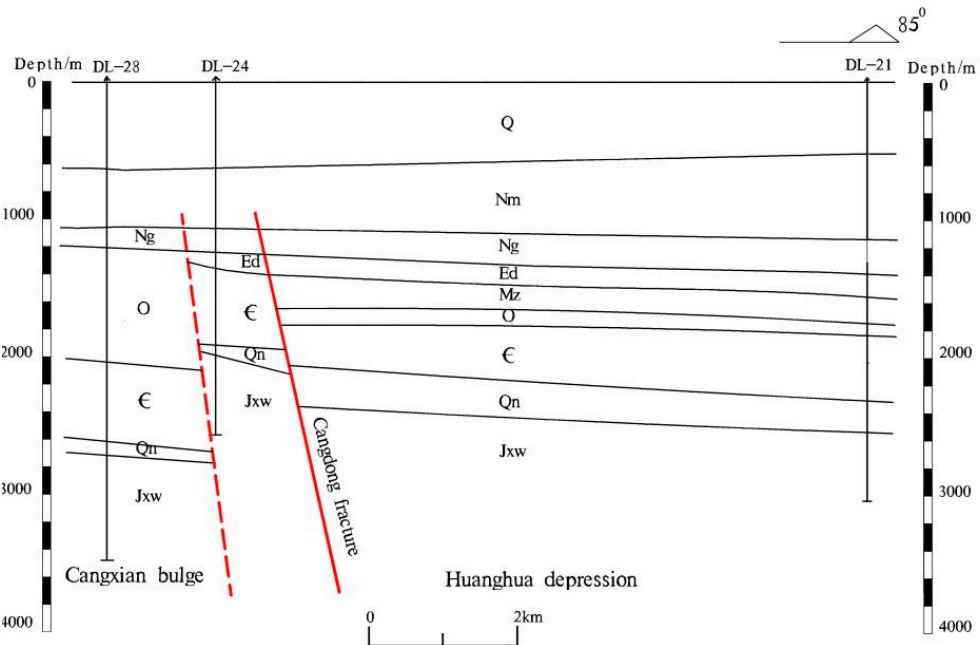

**Figure 2.** Geological section of Cangdong fracture.

Based on the lithology combination rule and geophysical prospecting data of exploration well TR21 [12] (Table 1), and combined with the study results of predecessors for the depositional environment of the Dongying formation in this area [13–15], the lithology characteristics of the three Dongying lithology sections are described as follows:

**Table 1.** The buried depth and thickness of the three Dongying lithology sections.

| Lithology Section | Buried Depth/m | Thickness/m |
|---|---|---|
| Dongying I segment | 1896–2044 | 148 |
| Dongying II segment | 2044–2214 | 170 |
| Dongying III segment | 2044–2214 | 148.98 |

Dongying I segment: the lithology is gray-green mudstone blending with gray-green powder sandstone; the upper part is a black, thin layer of shale stone.

Dongying II segment: the upper part is middle-coarse sandstone, thickness 12–15 m; the lower part is gray-green mudstone that blends with gray-green powder sandstone; the mudstone contains increasing amounts of debris; the main component is feldspar particles followed by quartz; the feldspar is weathered, with a poor sphericity and particle size 2–5 mm; the segment shows obvious anti-cycle deposition.

Dongying III segment: the bottom is gray-black/dark-brown mudstone; the upper part is gray-green powder sandstone blending with mudstone; the core samples of the segment of siltstone have horizontal bedding, with an average thickness of 5 m and particle size of about 5 mm, and the debris is an angular–sub-angular shape, with a poor rounding degree.

The Dongying formation deposition in this area is mainly a set of terrigenous clastic sediments with mudstone and siltstone, including feldspar, quartz and a small amount of debris. From a view of the sedimentary sequence, the debris particle sizes become gradually larger from the bottom to the top. The Dongying III segment at the bottom is black/dark brown mudstone; its upper part is gray-green mudstone, siltstone and increasingly clastic. The upper part of the Dongying II segment is middle-coarse sandstone, which is a sedimentary anti-cyclic sequence, and is an important feature of the delta sediments [16,17]. The profile curve of geophysical prospecting (Figure 3) can also reflect changes in sedimentary sequence. Combining the sediment characteristics, which are the low ratio of sand and mud, the more indicative marine fossils and so on, it can be seen that Dongying's terrigenous clastic deposits in this area are the delta sediments controlled by rivers [18,19].

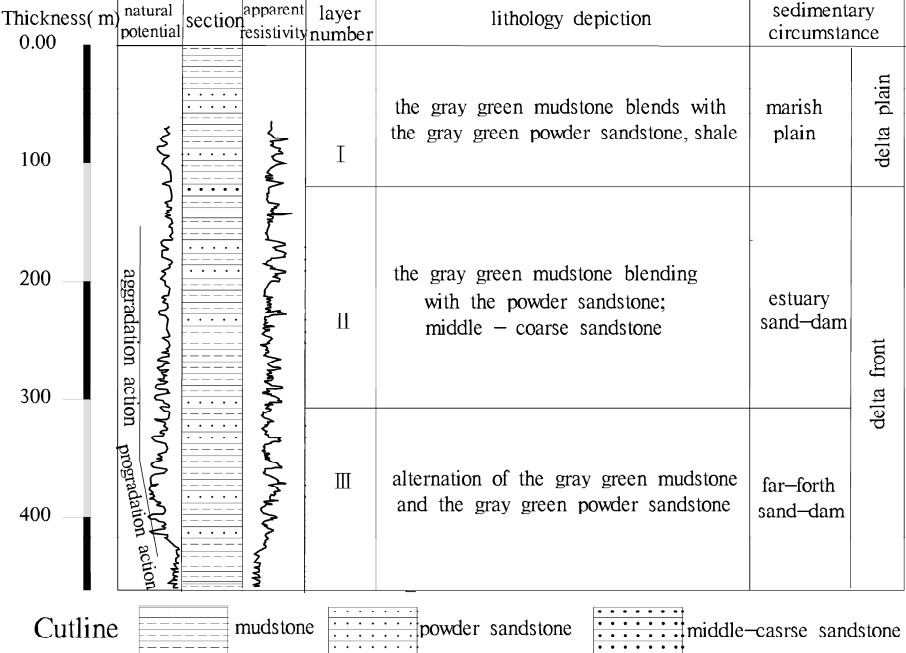

**Figure 3.** Dongying sedimentary faces analysis represented vertically.

According to the data of regional water level dynamic observation, the buried depth of the Guantao geothermal reservoir water level is between 66 and 94 m, and that of the Minghuazhen formation of Neogene is between 60 and 70 m. The water levels of the wells JN-02 and DL-16 of the Ordovician geothermal reservoir are about −60 m and −86 m, respectively, while the water level of well TR21 of the Dongying formation is about −36 m; the geothermal reservoir water level of the Dongying formation in this area is generally higher than that of the upper and lower geothermal reservoirs (Table 2). It can be seen from the water level dynamics that the geothermal fluid in the upper and lower reservoirs cannot supply the Dongying formation through the Cangdong fracture.

**Table 2.** Correlation of water column height at the same geothermal reservoir depth located at the east and west plate of the Cangdong fracture.

| The Well Number (Geothemal Reservoirs) | Converted to 40-Degree Water Column Height/m | Converted to Reservoir Temperature Water Column Height/m |
|---|---|---|
| DL-16 (O) in the west plate | −86 | −49.1 |
| DL-19 (Jxw) in the west plate | −97.6 | −49.3 |
| TR21 (Ed) in the east plate | −36 | +5.49 |
| TG-11 (Ng) in the east plate | −40 | −13.93 |

*2.2. Sampling and Analysis*

The development of Dongying's geothermal resources in the study area is still in its initial stages. Only two Dongying geothermal wells have been drilled so far. One of the wells is located to the southwest of the Xiangluowan Plaza, Binhai New Area. This geothermal well (TG-33) was transformed from an oil exploration well. The pumping test showed that the maximum water output of the single well was about 71.85 m$^3$/h, and the water outlet temperature was 85 °C, which was mainly used for heating. The other well (TR21) is the project exploration hole, located at Bincheng Longda Investment and Development Co., Ltd., Binhai New Area, Tian Jin, China. Its stable outlet water temperature is 73 °C, the water volume is about 40 m$^3$/h and only for bathing, swimming and mineral water development.

Geothermal fluid samples of the study area were collected between 2017 and 2022. In order to analyze the hydraulic relationship between the Cangdong fracture and the Dongying formation, sampling points were set up for different geothermal reservoirs in the Cangdong bulge area and Huanghua depression area on both sides of the Cangdong fracture. Additionally, environmental isotopes and radionuclide isotopes were also tested. A total of 21 water samples were collected, including 13 water samples of Minghuazhen geothermal fluid, 9 water samples of Guantao geothermal fluid, 2 water samples of Dongying geothermal fluid and 4 water samples of Ordovician geothermal fluid.

Each water sample was collected in a polyethylene bottle, which was rinsed three times with deionized water before sampling. The water sample for anion analysis was filtered through a 0.45 μm microporous membrane and stored in bottles. The sample used for cation analysis was also filtered and acidified with 6 N HNO$_3$ until pH < 2 and sealed for storage [20,21]. The water samples for hydrogen and oxygen isotope analysis were unfiltered and stored directly in polyethylene sampling bottles. In the field, a portable multiparameter (Hach HD40Q, Loveland, CO, USA) was used to determine the temperature (°C), pH, Total Dissolved Solids (TDS), Oxidation-Reduction Potential (ORP) and electrical conductivity [22].

Instruments used were Horiba Water Checker (Model U-10), Lovibond CM-21 Tintometer and Thermo Elemental M-Series Atomic Absorption Spectrometer (AAS). These instruments were calibrated before use. Quality checks were also performed on the instruments by checking the absorbance after every ten sample runs; tools and work surfaces were carefully cleaned for each sample. Minimum of triplicate readings were taken to check

precision of the analytical method and instrument. The pH, conductivity and turbidity were measured using Horiba Water Checker (Model U-10) after calibrating the instrument with the standard Horiba solution. Total dissolved solid (TDS) was measured with a Lovibond CM-21 Tintometer. Sulphate determination was by the turbidimetric method. Phosphate was determined using the stannous chloride method. Nitrate measurement was by the Brucine method. Chloride was determined by the argentometric titration method. Total alkalinity determination involved the titration of 50 mL sample, containing 5 drops methyl orange indicator with 0.02 N $H_2SO_4$ solution. The EDTA titration method was used in determining total hardness. The EDTA titration method was used in determining calcium. Magnesium concentration was determined. Sodium (Na) and potassium (K) concentrations in the water were measured using Thermo Elemental M-Series Atomic Absorption Spectrometer (AAS).

### 2.3. Data Processing and Analysis

The chemical composition of the geothermal fluid is formed by long-term exchange with the surrounding rock; the chemical characteristics of geothermal fluids in the Dongying formation of the Huanghua depression are affected by many factors, such as lithology, stratigraphic structure, distance between recharge and drainage, fracture structure and so on [23–25]. The chemical composition of the geothermal fluid is shown in Table 3.

**Table 3.** Major chemical composition table of Dongying geothermal fluid.

| Component | Concentration /mgL$^{-1}$ | | Component | Concentration /mgL$^{-1}$ | | Component | Concentration /mgL$^{-1}$ | |
|---|---|---|---|---|---|---|---|---|
| | TR21 | TG-33 | | TR21 | TG-33 | | TR21 | TG-33 |
| $Na^+$ | 977.3 | 1188.0 | $SO_4^{2-}$ | 10 | 39.2 | $S^{2-}$ | - | 0.61 |
| $K^+$ | 7.2 | 15.2 | $HCO_3^-$ | 845.1 | 839.0 | Soluble $SiO_2$ | 53.0 | 57.5 |
| $Ca^{2+}$ | 6.8 | 9.1 | $CO_3^{2-}$ | 12 | 24.0 | Dissociation $CO_2$ | 0.0 | 0.0 |
| $Mg^{2+}$ | 0.6 | 1.0 | $Cl^-$ | 1081.2 | 1293.9 | $COD_{cr}$ | 7.41 | 9.74 |
| $NH_4^+$ | 2.77 | 3.65 | $I^-$ | 2.00 | 1.75 | $HBO_2^-$ | 14.64 | 14.90 |
| $Fe^{2+}$ | 1.2 | 0.26 | $PO_4^{3-}$ | 0.04 | 0.08 | pH | 8.38 | 8.64 |
| $Fe^{3+}$ | 0.08 | <0.02 | $HBO_2^-$ | 14.63 | 14.90 | Total acidity | 0.0 | 0.0 |
| $Mn^{2+}$ | 0.02 | 0.01 | $NO_2^-$ | 0.005 | 0.007 | Total alkalinity | 713.1 | 728.2 |
| $Al^{3+}$ | 0.014 | 0.01 | $F^-$ | 5.0 | 3.96 | Total hardness | 19.5 | 26.5 |
| $Zn^{2+}$ | <0.02 | | $Br^-$ | 1.2 | 4.5 | A substance of solid shape | 2570.6 | 3047.4 |
| $Cu^{2+}$ | <0.02 | | $NO_3^-$ | 5.9 | 9.47 | TDS | 2993.2 | 3466.9 |

This study uses the Entropy-weighted water quality index (EWQI) to describe the water quality characteristics of the study area. As a convenient and effective water quality assessment method, the water quality index has been widely used by scholars all over the world [26–28]. Step 1 is assigning an entropy weight to each parameter. Step 2 is assigning a quality rating scale for each parameter. Step 3 is the calculation of the water quality index through the composite index method.

To assess the health risk of Dongying geothermal fluid, this study quantified the health risks from dermal contact using the empirical model proposed by the United States Environmental Protection Agency [28]. The detailed calculation process can be found in Table 4. The water quality result of Dongying fluid is poor.

**Table 4.** Characteristic elements and trace chemical components table of Dongying geothermal fluid.

| Component | Concentration /mgL$^{-1}$ | | Component | Concentration /mgL$^{-1}$ | | Component | Concentration /mgL$^{-1}$ | |
|---|---|---|---|---|---|---|---|---|
| | TR21 | TG-33 | | TR21 | TG-33 | | TR21 | TG-33 |
| $Hg^+$ | <0.0001 | <0.0001 | $Cd^{2+}$ | <0.001 | <0.001 | Se | <0.001 | <0.001 |
| TCr | 0.007 | 0.006 | Sr | 0.68 | 1.02 | Li | 0.118 | 0.201 |
| $Cr^{6+}$ | <0.004 | <0.004 | Ba | 0.417 | 0.564 | $CN^-$ | <0.001 | <0.001 |
| $As^{3+}$ | 0.003 | 0.012 | Ni | <0.01 | <0.01 | $H_2SiO_3$ | 53.0 | 74.8 |
| $Pb^{2+}$ | <0.01 | <0.01 | Ag | <0.05 | <0.01 | phenolic | 0.027 | 0.4 |

The main chemical composition of the geothermal fluid in each geothermal reservoir is shown in Table 5. The chemical characteristics between Dongying geothermal fluid and seawater is shown in Table 6.

**Table 5.** Correlation table of main chemical composition of the geothermal fluid in each geothermal reservoir.

| Geothermal Reservoir | Construction Location | The Well Number | TDS /mgL$^{-1}$ | Na$^+$ /mgL$^{-1}$ | Ca$^{2+}$ /mgL$^{-1}$ | Cl$^-$ /mgL$^{-1}$ | HCO$_3^-$ /mgL$^{-1}$ |
|---|---|---|---|---|---|---|---|
| Nm | Huanghua depression | TG-13 | 919.7 | 250.7 | 3.6 | 33.7 | 546.1 |
| | | TG-08 | 994 | 270.2 | 4.5 | 70.9 | 555.3 |
| | Cangxian bulge | DL-07 | 1525.5 | 471.2 | 9.3 | 315.5 | 463.8 |
| | | JN-01 | 1712.9 | 552 | 12.8 | 407.7 | 402.7 |
| Ng | Huanghua depression | TG-29 | 1759.2 | 549.4 | 12 | 374 | 518.7 |
| | | TG-01 | 1851.3 | 564.2 | 8.8 | 418.3 | 540 |
| | | TG-18 | 1684.7 | 514.1 | 11.1 | 340.3 | 543.1 |
| | | TG-23 | 1630.9 | 501.1 | 11.1 | 351 | 472.9 |
| | Cangxian bulge | DL-10 | 1752.2 | 502.4 | 28.5 | 374 | 543.1 |
| | | DL-25 | 1765.3 | 460.9 | 38.8 | 388.2 | 472.9 |
| Ed | Huanghua depression | TR21 | 2993.2 | 977.3 | 6.8 | 1081.2 | 845.1 |
| | | TG-33 | 3466.9 | 1188 | 9.1 | 1293.9 | 839 |
| O | Cangxian bulge | DL-16 | 1829.6 | 474.7 | 45.4 | 391.7 | 500.4 |
| | | DL-35 | 1752.7 | 466.3 | 40 | 386.4 | 497.3 |
| | | JN-02 | 1737.9 | 500 | 29.3 | 418.3 | 482.1 |

The isotopic composition of geothermal fluid depends on the isotopic composition of recharge water and its underground circulation process. The isotopic composition of groundwater without isotope exchange is consistent with that of recharge water, such as exchange with the surrounding rock (water–rock reaction); the isotopic composition of groundwater will change in different degrees [29]. The $\delta D$ value of geothermal fluid is relatively stable, but the $\delta^{18}O$ value varies, which is mainly the result of water–rock interaction. According to the composition of the hydrogen and oxygen isotopes of the samples, details of the recharge source, the recharge elevation and the interaction degree with the wall rock of the geothermal fluid can be inferred. The calculation steps and results are shown in Table 7.

**Table 6.** Correlation table of chemical characteristics between Dongying geothermal fluid and seawater.

| Component | Average Value of Dongying Geothermal Fluid/mgL$^{-1}$ | Average Value of Seawater/mgL$^{-1}$ |
|:---:|:---:|:---:|
| Na$^+$ | 1072.25 | 10648 |
| Ca$^{2+}$ | 8.1 | 420 |
| Mg$^{2+}$ | 0.95 | 1317 |
| Cl$^-$ | 1147.7 | 19324 |
| SO$_4{}^{2-}$ | 23.05 | 2688 |
| HCO$_3{}^-$ | 834.45 | 150 |
| I$^-$ | 1.375 | 0.06 |
| F$^-$ | 4.23 | 1.4 |
| Br$^-$ | 2.95 | 65 |
| TDS | 3164.7 | 35000 |
| *rNa/rCl* | 1.45 | 0.85 |
| *Cl/Br* | 501.45 | 292 |

**Table 7.** Statistics table of the hydrogen and oxygen isotope data of the geothermal fluid in each geothermal reservoir.

| Geothermal Reservoir | Construction Location | The Well Number | $\delta^{18}$O (‰) | $\delta$D (‰) |
|:---:|:---:|:---:|:---:|:---:|
| Nm | Huanghua depression | TG-13 | −9.6 | −71 |
|  | Cangxian bulge | JN-06 | −8.07 | −60.43 |
| Ng | Huanghua depression | TG-28 | −9.4 | −71 |
|  |  | TG-23 | −9.1 | −72 |
|  | Cangxian bulge | JN-03 | −9.36 | −61.19 |
| Ed | Huanghua depression | TR21 | −7.4 | −56 |
| O | Cangxian bulge | DL-16 | −8.7 | −72 |
|  |  | JN-02 | −9.8 | −81 |

Each radionuclide has a fixed decay rate or probability, independent of factors such as temperature, pressure, electromagnetism and centrifugation; it is also independent of the geological history, age and chemical state of the isotopes [30]. The geothermal water age is calculated by measuring C14. The abundance of C14 is measured by the conventional radioactive decay technique. The calculation steps and results are shown in Table 8. Radioactive decay obeys the following decay laws:

$$N_t = N_0 e^{-\lambda t} \tag{1}$$

$$\lambda = \ln 2 / T \tag{2}$$

**Table 8.** Statistics table of the age data measure of the geothermal fluid in each geothermal reservoir.

| Geothermal Reservoir | Construction Location | Statistics of Well Amount (Holes) | Minimum Age /ka | Maximum Age /ka | Average Age /ka |
|---|---|---|---|---|---|
| Nm | Cangxian bulge | 11 | $19.5 \pm 0.24$ | $31.33 \pm 1.15$ | 23.21 |
| | Huanghua depression | 2 | $21.55 \pm 0.46$ | $27.40 \pm 0.46$ | 24.48 |
| Ng | Cangxian bulge | 5 | $16.91 \pm 0.20$ | $24.10 \pm 1.40$ | 21.49 |
| | Huanghua depression | 4 | $24.01 \pm 0.53$ | $25.56 \pm 0.40$ | 24.80 |
| Ed | Huanghua depression | 2 | $31.549 \pm 0.58$ | $31.549 \pm 0.58$ | 31.55 |
| O | Cangxian bulge | 4 | $15.421 \pm 0.542$ | $29.451 \pm 1.552$ | 21.52 |

From Formula (1) can be exported:

$$t = \frac{1}{\lambda} \ln \frac{N_0}{N_t} \tag{3}$$

It is known that the half-life of $^{14}$C is 5730 a. The $^{14}$C dating age formula is derived from Formula (3) of the radioactive decay law:

$$t = 82671 \ln \frac{N_0}{N_t} \tag{4}$$

where $N_t$ (Tu) is the residual concentration after $t$-time decay; $N_0$ (Tu) is the concentration at time zero; $\lambda$ is the decay constant of a radioactive element; T (a) is the half-value period; $t$ (a) is the decay time from time zero to the determination time of the sample.

## 3. Results and Discussion

### 3.1. Hydrochemical Characteristics

The Dongying geothermal wells of the study area include well TR21 and well TG-33. The water intake section of well TR21 is 1952.67–2294.88 m, the water outlet temperature is 73 °C and the pH is 8.38. The water intake section of well TG-33 is 2199–2454 m, the water outlet temperature is 85 °C and the pH is 8.64. The chemical composition of Dongying geothermal fluids in the area is shown in Table 3.

According to the water quality analysis results of the two Dongying geothermal wells in the study area, the main cation of Dongying geothermal fluid is $Na^+$ and the main anions are $Cl^-$ and $HCO_3^-$; the TDS is high, about 2800–4400 mg/L, indicating that the chemical characteristics of Dongying geothermal fluid in this area are mainly alkali and strong acid ions.

The $rNa/rCl$ (the milligram equivalent ratio of $Na^+$ and $Cl^-$ in groundwater), which is closely related to the depositional environment and history of transgression, is also a basic index to judge the deterioration degree of natural water and the intensity of water alternating action. The rNa/rCl of well TR21 and well TG-33 are 1.47 and 1.42, respectively. The rNa/rCl of the Dongying geothermal reservoirs were about 1.4 > 1. It is the syngenetic sedimentary water of the lacustrine river delta, which has typical characteristics of the dissolved and filtered water. According to Shukarev classification, the main hydrochemical type of Dongying geothermal fluid in the study area is the $CL \cdot HCO_3$-Na type.

The Dongying geothermal fluid in the study area is characterized by a high molar content of $HCO_3^-$, which is higher than the total molar content of calcium and magnesium ions. This indicates that there is excess $HCO_3^-$ binding to $Na^+$ in the hot fluid, which is due to exchange between the calcium and magnesium ions of the geothermal fluid and the sodium ions adsorbed by the wall rocks. According to the data of the geothermal wells, the

total iron content is 0.28–1.28 mg/L, which exceeds the limit of drinking water, 0.3 mg/L, and the $Fe^{2+}$ content is much higher than $Fe^{3+}$. After contact with air, $Fe^{2+}$ reacts with oxygen and the water will turn rusty red.

The metaboric acid ($HBO_2^-$) content of the geothermal fluid in the Tianjin area is generally high. The metaboric acid ($HBO_2^-$) content of the Dongying geothermal fluid is also high in the study area, at about 15 mg/L, which is about ten times as much as in quaternary cold water. The fluoride ion content of the geothermal fluid in Tianjin is generally high, and exceeds the standard of drinking water. The highest content of fluoride ions is 18.6 mg/L and generally 6–11 mg/L. However, the fluoride ion content of Dongying geothermal fluid in the study area is low, ranging from 3 to 5 mg/L.

Any materials present in the Earth's crust may dissolve in the geothermal fluid and affect the mineral composition of the geothermal fluid. Medium and trace metal elements were detected in Tianjin geothermal fluid, including Li, Sr, Ba, Ni, Ag, Co, Mo, Se, $Fe^{2+}$, $Fe^{3+}$, $Cu^{2+}$, $Al^{3+}$, $Mn^{2+}$, $Zn^{2+}$, $Hg^{2+}$, TCr, $Cr^{6+}$, $As^{3+}$, $Pb^{2+}$ and $Cd^{2+}$. The heavy metals that can pollute the environment and have obvious biological toxicity are mainly Hg, Sr, Pb, Cr and As, but also include general toxic heavy metals such as Zn, Cu, Ni, Co, SE, etc. According to the geothermal fluid quality analysis of the study area (Table 4), the $Hg^{2+}$, TCr, $Cr^{6+}$, $Pb^{2+}$, $Cd^{2+}$, $Zn^{2+}$, $Cu^{2+}$ and Ni contents of Dongying geothermal fluid were all superior to the class II groundwater quality standard, and some of the indexes reached class I groundwater quality standard. The $As^{3+}$ and Ba contents were somewhat high; they could only meet the class III groundwater quality standard. The raw water of Dongying geothermal fluid in the study area will not cause heavy metal pollution to the environment. However, the phenol content of the geothermal fluid is seriously over the legal limit, and phenol is harmful to the human body. Therefore, the $As^{3+}$, Ba and phenolic components of Dongying geothermal fluid should be treated properly according to their utilization.

### 3.2. The Analysis of Main Chemical Composition

The chemical composition of the geothermal fluid is formed by long-term exchange with the surrounding rock. The chemical characteristics of Dongying geothermal fluid in the Huanghua depression are affected by many factors, such as lithology, stratigraphic structure, distance between recharge and drainage, fault structure and so on. The relationship between salinity and the burial depth of Paleogene geothermal fluid in the Huanghua depression is very close, mainly manifested in the increase in burial depth and the increase in the salinity of the geothermal fluid. The relationship is nearly a linear relationship between the total dissolved solid (TDS) and burial depth (Figure 4) [23]. The amplitude increase is slowed down at depth, mainly because the geothermal water recharge cycle at the deeper part is weaker than that at the upper part, and the salinity of geothermal fluids is reduced due to the dehydration of clay minerals in the overpressure zone [31].

The relationship between the TDS of Dongying geothermal fluid and the burial depth is similar to regional characteristics. In the Dongying geothermal well TR21, the TDS of the geothermal fluid is 2993.2 mg/L, while in the geothermal well TG-33 the TDS of the geothermal fluid is 3466.9 mg/L. According to the nearly linear relationship between the total dissolved solid and burial depth shown in Figure 4, the TDS of TG-33, which is the same water intake depth as well TR21, can be estimated to be about 2700 mg/L. Based on the characteristics of the TDS of the geothermal fluid of the two existing wells at the same reservoir depth, and combined with regional variation characteristics, the TDS of Dongying geothermal fluid is the lowest near the intersection of the Haihe fracture and the Cangdong fracture, increasing from the intersection, respectively, to the southwest and northeast in two directions. The result shows that the geothermal fluid runoff is quite obvious near the intersection of the Haihe fracture and the Cangdong fracture, and the geothermal fluid is conducted along the Haihe fracture.

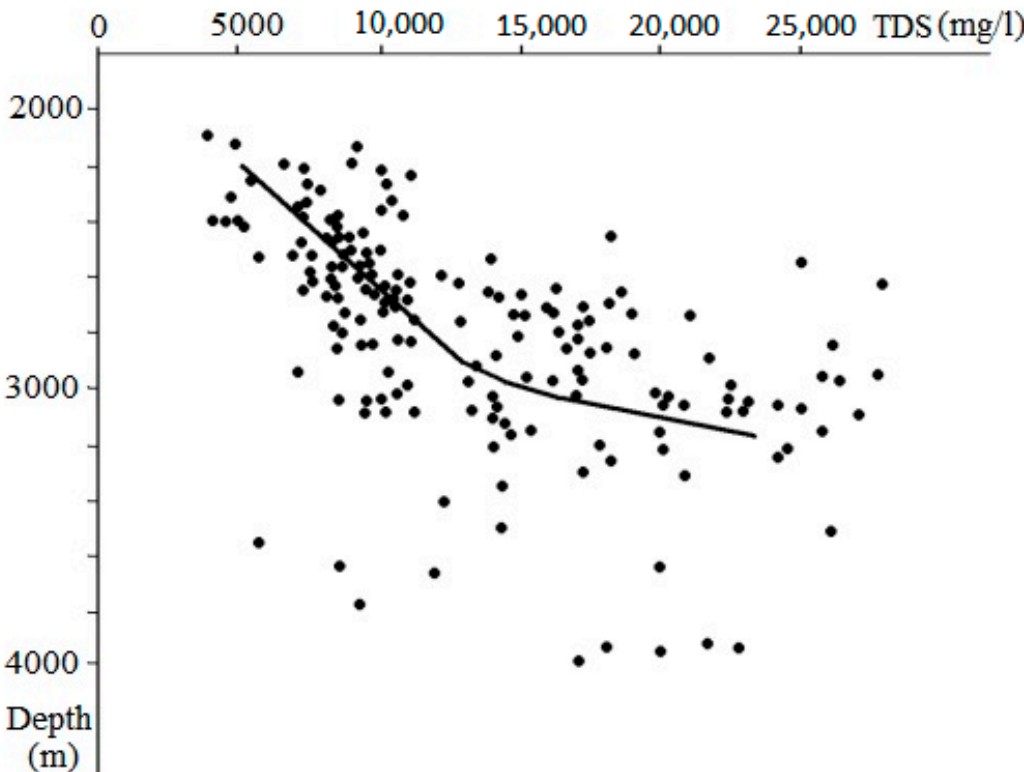

**Figure 4.** Relationship between the TDS and the burial depth of Paleogene fluid in the Huanghua depression.

The main chemical components of Dongying geothermal fluid in the study area have great vertical changes within its overlying and underlying thermal reservoirs [32,33] (Table 5). It can be seen that the hydraulic connection between Dongying geothermal fluid and its overlying and underlying geothermal reservoirs is weak. Compared with the main chemical component contents of Ordovician, Guantao formation and Minghuazhen geothermal fluid in the west of the Cangdong fracture there is a great difference, which also shows that there is no obvious hydraulic connection between the geothermal reservoirs in the west of the Cangdong fracture and the Dongying formation of geothermal reservoirs in the study area.

The chemical composition of the Dongying geothermal fluid in the study area was compared with that of the seawater (Table 6). There was a significant difference between two. The Na+ content and the TDS of the geothermal fluid were 10 times lower than the mean value of the seawater; the contents of $Mg^{2+}$ and $Ca^{2+}$ were 2–3 orders of magnitude different from the sea water. The content of $Cl^-$ was about 15 times lower than the average value of the sea water, the contents of $I^-$ and $F^-$ were about 3 times higher than the average value of the sea water and the content of $Br^-$ was about 22 times lower than the average value of the sea water. In addition, the *rNa/rCl* and *Cl/Br* of the Dongying geothermal fluid were 1.45 and 501.45, respectively, which were higher than those of seawater, indicating Dongying geothermal fluid is of a non-seawater origin.

*3.3. The Analysis of the Hydrogen and Oxygen Isotope Data*

Based on the analysis of the hydrogen and oxygen isotope data [34–36] (Table 7), it can be seen that the hydrogen and oxygen isotope values of the Minghuazhen formation and Guantao geothermal fluid are similar, and they are significantly different from the Dongying geothermal fluid. Moreover, the δD value of the geothermal fluid is relatively stable, but the $δ^{18}O$ value is various, which is mainly the result of water–rock interaction. Therefore, this shows that there are differences in the supply source between the Dongying

geothermal reservoir and the upper and lower geothermal reservoirs in the study area, which also shows that there is no obvious hydraulic connection between the geothermal reservoirs in the west of the Cangdong fracture and the Dongying formation geothermal reservoirs in the study area.

*3.4. The Analysis of the Radionuclide Isotope Data*

The statistical data of the radionuclide isotopic dating for different geothermal reservoirs and different tectonic units were analyzed [37,38] (Table 8). The geothermal fluid age characteristics were as follows: ① The geothermal fluid age of the Dongying formation in this area is similar to that of Well JN-01 and well JN-02 of the Minghuazhen formation in the Cangxian bulge district of Tianjin; ② The geothermal fluid age of the Dongying formation in the general survey area is older than that of the overlying strata in the Huanghua depression district of Tianjin; ③ The age of the bulge area is younger than that of the depression area for the pore geothermal fluid; the geothermal fluid ages between the Dongying formation and the upper and lower geothermal reservoirs are significantly different, indicating that there is no obvious hydraulic connection between the Dongying formation and the upper and lower thermal reservoirs.

## 4. Conclusions

Through the analysis of the hydrochemical information, the hydrogen and oxygen isotope data and the radionuclide isotope data between the Dongying formation geothermal reservoir and the overlying and underlying geothermal reservoirs of the Tianjin coastal regions, the important parameters of geothermal hydraulic connection can be obtained in detail, and the method is shown to be effective. This method partly addresses the lack of research into the geothermal fluid circulation of deep geothermal sedimentary aquifers. The main conclusions are as follows:

(1) The main chemical composition, the hydrogen and oxygen isotope data and the age values of geothermal fluid are significantly different between the Dongying geothermal reservoir and the upper and lower geothermal reservoirs.

(2) The hydraulic conductivity in the Cangdong fracture and the Dongying formation geothermal reservoirs in this area is weak; there is only a certain hydraulic conductivity along the Haihe fracture at the intersection of the Haihe fracture and the Cangdong fracture.

(3) There is no obvious hydraulic connection between the Dongying formation geothermal reservoir and the overlying and underlying geothermal reservoirs of the Tianjin coastal regions.

**Author Contributions:** J.L. conducted the investigation and data collection, developed the suitable methodology and was involved with the writing of the manuscript. S.H. provided project administration and was involved in the writing and editing of the manuscript. F.Y. and D.Y. helped in methodology selection, and participated in editing the earlier versions of the manuscript. All authors have read and agreed to the published version of the manuscript.

**Funding:** This research was funded by the China Geological Survey's project: Investigation and monitoring of hydrogeology and water resources in the middleland lower reaches of the Yellow River basin (DD20221754), and Investigation and evaluation of dry-hot rock resources in eastern China (DD20221680).

**Institutional Review Board Statement:** Not applicable.

**Informed Consent Statement:** Not applicable.

**Data Availability Statement:** Not applicable.

**Conflicts of Interest:** The authors declare no conflict of interest.

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
