# Peer review of "The Hydraulic Connection Analysis of Dongying Geothermal Fluid Using Hydrochemical Information and Isotope Data in Tianjin Coastal Regions"

_water, doi:10.3390/w15061235_

Round 1

Reviewer 1 Report

1.       Please check the spacing between words and sentences in the entire text.

2.       I suggest adding nomenclature to describe the abbreviation and symbol of equation.

Point 1: Introduction

1.       I suggest mentioning previous research data related to the topic of research to find out the novelty of this research.

Point 2: Materials and Methods

1.       Line 76-78 page 2: Please adjust font size in the text.

2.       Line 78 page 2: Please adjust the minimum spacing between the text on line 78 and Figure 1 (I think it is too wide). I suggest moving the explanation regarding Figure 2 to the next page (in paragraph that is not too far from the figure).

3.       Line 84 page 3: Please describe the abbreviation of NNE and SEE in advance. Then, the author can use the term in next sentence. It is also similar to other terms.

4.       Line 105 page 4: I suggest showing the explanation of the lithology characteristics of the three Dongying lithology sections in the table.

5.       Figure 3 page 5: Please adjust the font in the figure (make it clearer).

6.       Line 168 page 6: Please check subscript format in “HNO3”.

7.       Equation (1) and (2) page 6: Please adjust font size in equation labels (1) and (2). Please see equation label of (3) and so on.

Point 3: Results and Discussion

1.       Please add references to support/compare the result with previous research.

2.       Line 280 page 9: Please check Figure 3 which you mentioned in the sentence " According to the nearly linear relationship between the total dissolved solid and burial depth shown in Figure 3,.…". I think it should be Figure 4.

3.       Table 4 page 9: I suggest fixing Table 4 into one page. It is also similar to Table 6.

4.       Line 327 page 11: I suggest showing the explanation of geothermal fluid age characteristics in the table.

Point 4: References

1.       I suggest updating references for at least the last 5 years.

2.       Please check consistency reference format such as capital letter format in article and journal title, etc. (Please check references of [20], [26], etc.).

Reviewer 2 Report

The present paper reports the hydrological characteristics and identify the hydrological connection between geothermal reservoirs. The study is interesting and should have practical value to some extent. However, the paper is not organized very well and substantial modification is needed. The English should be improved before consideration for publishing. The detailed comments are provided below:

Please use the half-angle symbol throughout the paper, and check if subscripts have been used for ions.

Line 20: the authors should have a clear distinguishment between "stable" and "environmental" isotopes. What are the differences between them?

In the abstract, the authors have presented too much space to describe the background. Please highlight the main methodology and main findings.

Lines 56-57: this sentence has grammar error. Please re-write it.

In the introduction, literature review should be described, including the previous studies about the hydrological connection, hydrological characteristics or hydrogeological exploration in you study area.

Line 64: the correct expression should be "of great significance".

Figure 1: it is suggested to note the location of your study area in a Chinese map.

Table 1: the caption should be re-written.

Line 180: How the EWQI is calculated should be described.

Line 182: where is Table 2 & 4? I cannot see a supplementary file.

Lines 193-195: a reference should be added for this statement.

Equation (3) is not present clearly. please re-write it.

In the methodology section, you should mention the measurements of hydrochemical and isotopic compositions.

Line 322: "weak" instead of "small".

Section 3.3: You cannot determine the connection just based on 1-time isotopic composition. Relationship between hydrogen and oxygen isotopes in geothermal water is recommended to further determine the connection between formations.

Reviewer 3 Report

Paper is very good. I suggest to publish that scientific paper.

Round 2

Reviewer 2 Report

The authors have improved the paper greatly. I agree it can be published in Water.

Author Response

Thank you very much!